# Analysis of the Destructive Effect of the *Halyomorpha halys* Saliva on Tomato by Computer Tomographical Imaging and Antioxidant Capacity Measurement

**DOI:** 10.3390/biology11071070

**Published:** 2022-07-18

**Authors:** Sándor Keszthelyi, Szilvia Gibicsár, Ildikó Jócsák, Dániel Fajtai, Tamás Donkó

**Affiliations:** 1Department of Agronomy, Institute of Agronomy, Hungarian University of Agriculture and Life Sciences, Kaposvár Campus, S. Guba Str. 40, H-7400 Kaposvar, Hungary; szilvia.gibicsar@gmail.com (S.G.); jocsak.ildiko@uni-mate.hu (I.J.); 2Medicops Nonprofit Ltd., S. Guba Str. 40, H-7400 Kaposvar, Hungary; daniel.fajtai@sic.medicopus.hu (D.F.); donko.tamas@sic.medicopus.hu (T.D.)

**Keywords:** brown marmorated stink bug, nondestructive analysis, plant impairment, tissue destruction, tomato

## Abstract

**Simple Summary:**

*Halyomorpha halys* is a devastating agricultural pest, and digestion starts with an extra-oral phase as the injury is inflicted by injected saliva enzymes into the plant tissues. We carried out a noninvasive imaging assay assisted by computer tomography (CT) of through damaged tomatoes caused by *H. halys*. It was intended to support the imaging results by further laboratory analytical approaches, such as the antioxidant capacity, which have been used as a stress indicator. Our results confirmed that the noninvasive approach may provide new data for the cognition of the degree of damage induced by this important pest. The important finding is the proof of escalation of the lesions as a function of bug number and the exposure time. Volume and structure deviation in tomatoes were justified by the antioxidant activity. In summary, our method can be suitable for the qualitative inspection of tomato items.

**Abstract:**

Qualitative and quantitative parameters of tomatoes are impaired by *Halyomorpha halys* Stål (Hemiptera: Pentatomidae), which cause severe economic losses worldwide. Our aims were to assess *H. halys*-induced tissue damage in tomatoes via computer tomography and to confirm the results of imaging obtained by analytical methods. Our examination confirmed the intensification of the change in the inner structure of damaged tomatoes as a function of time. The tendency of this destruction triggered by bug saliva grew exponentially from the exocarp layer to the inner placenta. The destruction of the plant tissue was aggravated by an increase in the number of bugs, as it was unequivocally evinced by the shell thickness assays. The results of the assessment of the antioxidant capacity of tomato mesocarp showed a distinct decrease in the antioxidant capacity of the samples obtained from *H. halys*-infested tomatoes. This indicates that the ferric-reducing antioxidant power value was related to the degradation processes of the mesocarp tissue in tomato fruit caused by the watery saliva released by *H. halys*. The presented experimental method can be suitable for the qualitative control of the vegetable items intended for trade, which can help for the isolation of tomatoes damaged by bugs immediately after harvest.

## 1. Introduction

Tomato is one of the most significant vegetable crops in the world, with a continuously increasing production area [1]. The production is endangered by true bugs, all over the world. Thus, *Halyomorpha halys* Stål (*Hemiptera: Pentatomidae*) possesses huge economic importance.

This pest is native to East Asia [2]. The earliest records of the pest in North America date back to 2001 and that in Europe to 2004 [3,4]. The spread, feeding, and reproduction of the species are highly linked to anthropogenic activities [5,6,7,8]. The developmental duration depends on photoperiodism and temperature. After hatching, the first instars stay in the proximity of the eggshells until molting, afterward they disperse to feed on host plants [9]. The insects overwinter as adults; the climatic conditions determine the number of annual generations [10]. It is considered a polyphagous pest, having more than 100 reported host plants [11]. The most important host plants include pepper, bean, pear, apple, peach, grapes, soybean, tomato, and corn [8,11,12,13,14].

The digestion of the true bugs (*Hemiptera*) starts with an extra-oral phase, as the injury is inflicted by injected saliva enzymes into the plant tissues [15,16,17]. The enzyme composition of the saliva depends on the feeding behavior of the particular species [18]. During the plant response, secondary compounds are synthesized, and the modified cell structure of the stressed tissue appears [15]. Similar to all other members of the Heteroptera suborder, *H. halys* feeds on plant parts being particularly rich in nutrients and energy. As compared to true bugs, their feeding behavior is more explorative resulting in more substantial damage. They can cause severe necrosis, various wound responses, and disturbances in the plant hormonal system; the inflicted damage may even result in death of the plant [15]. The phytophagous heteropteran bugs mainly specialized in consuming seeds and fruit crops; their saliva, therefore, contains cell-degrading enzymes. The feeding of the *H. halys* may cause discolorations, necrotic- and chlorotic patches, or deformed crops as the result of tissue damage [8,12].

Tissue structure changes upon decay processes affect the antioxidant capacities of fruit manner negatively [19]. The antioxidant capacity and lipid oxidation of tomato has been good indicator of quality [20]. However, there is no information on the effect of insect-caused damage as biotic stress on the antioxidant parameters of tomato or specifically stemming from *H. halys*.

Using digital imaging techniques, structural, and physiological alterations can be noninvasively studied. Previously, only destructive procedures based on traditional tissue excavation were available, such as sectioning, various tissue staining, or dissection methods [21]. The most important imaging techniques applied in pest and pest-induced damage diagnostics are U/S (ultrasound) X-ray imaging, CT (computed tomography), MRI (magnetic resonance imaging), CLSM (confocal laser scanning microscope), and IRT (infrared thermography), as well as the processing of the extracted data with RGB or grayscale imaging of the surfaces [22]. CT imaging is a well-known method in botanical research; mainly used for pest detection, pest morphology, or in the developmental and behavioral studies [23,24]. Studies examining pest damage in a noninvasive way are scarce [25].

Our study aimed to determine *H. halys*-induced tissue damage in tomatoes, based on CT imaging. We aimed to assess the effect of time and the number of feeding pests on the induced, destructive effect observed in tomato fruit. It was intended to support the imaging results by further laboratory analytical approaches, such as the antioxidant capacity, which have been used as a stress indicator. Our research may provide a new aspect on the degree of damage induced by this important pest.

## 2. Materials and Methods

### 2.1. Sampling and Experimental Setting

To determine the change in the volume and inner structure of tomatoes caused by *H. halys*, mixed-gender adults were collected from an insecticide-free environment. The collection was carried out in early September 2020 in the Kozármisleny area (Hungary, Baranya county, GPS coordinates: 46°01′46.81″ N 18°17′31.56″ E). The collected specimens were placed in an incubator until the beginning of the artificial infestation was carried out. The conditions set in the incubator (temperature: 25 °C and (15 L:9 D) photoperiodic parameter) mimicked the typical conditions in Hungary at that time of the year. In parallel, healthy, damage-free tomatoes were involved in the experiment, the diameters of which were uniformly 5–6 cm. At the beginning of experiment 1, 2, 5, and 8 bugs per treatment were placed on tomatoes in a transparent plastic box (20 × 15 × 10 cm). Each treatment was repeated 5 times. Subsequently, CT images were taken at 0, 3, 12, 24, 48, and 72 h after infestation to determine the change ensuing in the volume and inner tissue structure. Between each imaging, the plant samples with the pests were put back into the incubator under the same abiotic parameters (see above).

On the CT images, the volume changes and the extent of the destruction as well as the penetration depth of the salivary injected by *H. halys* were determined together with the tissue destructive tendencies caused by phytotoxic enzymes as a function of exposure times.

### 2.2. CT-Imaging and Post-Processing

The CT acquisition was carried out by a Siemens Somatom Definition AS+ (Siemens, Erlangen, Germany) using the settings as follows: tube voltage 100 kV, current 210 mAs, spiral data collection with pitch factor 0.7, and collimation 128 × 0.6 mm. The tomatoes placed in the box on the examination table were examined and the images were reconstructed with 153 mm field of view. The stored images in DICOM (Digital Imaging and Communications in Medicine) format were converted to NifTI (Neuroimaging Informatics Technology Initiative) format resulting in quasi isotropic voxel size (0.299 × 0.299 × 0.3 mm); then, every image processing step was performed in Python with open-source or custom-made software library and code. In the CT image of healthy tomato, the mesocarp and inner placenta region were well separated, typically voxels below −150 HU belong to the inner placenta region. A structural change was recorded for voxels below this HU value (threshold) in the mesocarp region (Figure 1a). To assess the spatial and temporal distribution of this phenomenon, a virtually undamaged mask was created for every tomato with further binary morphological operators to fill the grooves and holes related to structural changes while trying to maintain the normal surface curvatures. To describe the tissue properties of the tomato in different depths, we created 20 binary masks with iterative binary erosion of the undamaged mask, resulting in approximately 1 mm thick, nested shells. In each shell, a parameter was calculated—SRn—which describes the proportion of the voxels below the threshold in the nth shell for describing the tissue quality (Figure 1b).

### 2.3. Spectrophotometric Stress Assessment Assay

The mesocarp of tomato fruit was separated with a scalpel and homogenized with quartz sand in a cooled mortar. The resulting suspension was transferred to 0.015 L centrifuge tubes and centrifuged (10,000 *g*, 4 °C) for five minutes in a Hettich Mikro 220 R benchtop centrifuge (DJB Labcare Ltd., 20 Howard Way, Interchange Business Park, Newport Pagnell, UK). The supernatant was dispensed into 0.002 L Eppendorf tubes and used to measure total antioxidant capacity and lipid peroxidation.

Total antioxidant activity was measured by the modified assay of ferric reducing antioxidant power (FRAP) of Benzie and Strain [26]. The constituents of the FRAP reagent were the following: acetate buffer (300 mM pH 3.6), TPTZ (2,4,6-tripyridyl-s-triazine) 10 mM in 40 mM HCl, and FeCl_3_·6H_2_O (20 mM). The working FRAP reagent was prepared by mixing acetate buffer, TPTZ, and FeCl_3_·6H_2_O in a ratio of 10:1:1 at the time of use. The standard solution was 1000 µM ascorbic acid prepared freshly at the time of measurement.

To 0.0001 L of the supernatant was added 0.0029 L of FRAP reagent in 0.005 L screw cap centrifuge tubes, vortexed in a 37 °C water bath (Julabo ED-5M, JULABO GmbH, Seelbach, Germany) for four minutes, and the absorbance was measured at 593 nm against a blank with a BIORAD SmartSpec™ Plus spectrophotometer (Bio-Rad Ltd., 1000 Alfred Nobel Drive, Hercules, CA, USA). The FRAP values of the samples were determined in ascorbic acid (AA) equivalent (µg AA equivalent/mL extract) based on the ascorbic acid calibration curve. The results were calculated in ascorbic acid equivalents, as the averages of three independent measurements.

### 2.4. Statistical Analysis

In order to test the data derived from the volume and the tissue analysis (*n* ˃ 50), the Shapiro–Wilk test was employed. For the survey of the normal distribution of data (*p* < 0.05), the method of Ghasemi and Zahediasl [27] was used. The destructive effects of the injected bug saliva on the volume alteration of tomato were statistically analyzed by one-way ANOVA. The effects of the applied exposure times and the bug numbers as the independent factor on the tissue structure alteration (SR) as a dependent factor were statistically analyzed by two-way ANOVA. Mean values were separated using the Tukey (HSD) test at *p* ≤ 0.05.

The values derived from the antioxidant capacity assay were plotted as an average of three independent experiments and presented together with standard errors (±SE). The effect of the treatments on the measured variables was determined by one-way ANOVA (*p* ≤ 0.05), Duncan test, and SPSS 7.0 statistical program [28].

## 3. Results

### 3.1. Estimation of Volume Change by CT

The effects of *H. halys* sucking on the volume alteration of the experimental tomatoes are shown in Figure 2 and Figure 3.

These volume differences caused by pests between the damaged and intact samples were unequivocally proven by statistical analysis. The average volume decrease in the experimental tomatoes was 1.57% by the 72nd hour of the duration of the exposure. The distortion and volume decrease in the damaged fruit affected by the increase in the pest number were confirmed. It can be seen that the exponential type of volume decrease was generated by the increase in the number of sucking images. In line with preliminary expectations, the highest volume decrease was detected when the longest exposure time to eight pests was applied. The effect of the insect number on the volume decrease in the sucked tomatoes was also proven by statistical analysis.

The ascendant tendency of the volume decrease caused by *H. halys* depends on the exposure time that can be determined. The most spectacular change was attributable to the collective damage done by eight imagoes of the pest. The time dependency of the volume change could only be confirmed in the case of the highest individual number of pests applied. In contrast, the effect of the incremental exposure time on the volume change could not be justified in the tomato samples. The common impact of the analyzed two factors was not confirmed by the two-way ANOVA, either.

### 3.2. Determination of Inner Structure Alteration Using CT

Our examination confirmed the intensification of the damage occurring in the inner structure of tomatoes as a function of time. Nevertheless, after the first exposure time (3 h), the effect of the pest number on the tissue structure of tomato has not been statistically proven yet (*p* > 0.05). However, the tissue-destructive consequences of the mass alteration of injected salivary were justified by the statistical analysis in every case.

The exocarp/non-mesocarp ratio (SR) measured concentrically inward can be seen in Figure 4. It can be ascertained that the tendency of this value grows exponentially from the exocarp layer to the inner placenta region according to the natural inner tissue structure of tomatoes. The destruction of the plant tissue aggravated by the increase in bug number was unequivocally evinced by the shell thickness assays. The change in the tissue structure recorded as a function of time was not statistically justified in the case of samples containing one and two bugs, while the presence of lesions in different shells has shown differences.

Statistically verifiable differences were determined in relation to different exposure times of samples with higher pest numbers. In parallel, the tissue structures of different shells (SR) also differed in these samples.

Based on the graphs it can be established that the enzymatic degradation effect of the phytotoxic salivary has mostly expressed between shell 3 and 8 (SR3–SR8); thus, it appears that the saliva affected the middle parts of the mesocarp and vascular bundle regions 3–8 mm from the surface, on average (Figure 4). In none of the cases did the tissue-destroying effect extend to the inner regions of tomato containing the seeds, such as placenta or columella. This can be seen in the 72nd hour samples containing two bugs, but it is most pronounced in the 12th hour samples containing five and eight bugs. The most drastic tissue destruction was detected in the 72nd samples containing eight bugs.

### 3.3. Results of Stress Indicating Analytical Methods

The results of the antioxidant capacity of tomato mesocarp (Figure 5) show a distinct decrease of the samples extracted from *H. halys-*infested tomato fruit, as compared to the control. The rate of decrease was: 24%, for one bug; 26% for two bugs; 14% for five bugs; 19% for eight bugs. The statistical analysis revealed that the damage of tomato fruit caused by the bug lowered the FRAP values compared to that of the control fruit.

## 4. Discussion

Based on our investigation, it can be concluded, that the volume change- and tissue-destructive effects caused by *H. halys* on tomatoes can be studied by means of computed tomography. Some parallel research is available, which have examined the destructive effects of some phytophagous insects on the inner host tissues using computed tomography. A similar study was published by Farinha et al. [29], who evaluated seed content via more noninvasive approaches, which was expected to better characterize the damage caused by the western conifer seed bug (*Leptoglossus occidentalis* Heidemann, 1910). According to their results, the adults of *L. occidentalis* were capable of feeding on mature seeds by piercing the hard and thick coat, and useful results were obtained on the extent of internal tissue damage.

The invasive *H. halys* can trigger remarkable damage to several host plants during its establishment in a new habitat. According to Stahl et al.’s [30] results, *H. halys* adults can cause more external damage to pistachio than the native species, which can be explained by the high number of individuals. However, this pest does not cause more kernel necrosis (internal damage) than the native species do.

In contrast, tomato is more susceptible to serious injuries than stone fruit due to its soft tissue structure. Based on our examination, it can be established, that volume alteration coupled with the change in tissue structure occurs in the 72nd hour upon infestation. which is confirmed by CT imaging and analytical examination. However, a longer time is required for a measurable structure alteration caused by pests, as the remarkable changes can only be observed from the 72nd hour. The tissue-destructive consequences of the bug’s feeding can be explained by the presence of the enzymes injected by the bug into the plant tissue, which was confirmed by several former studies [31,32,33].

According to our measurements, the mean lesion depth is 3-8 mm from the exocarp, which coincides with the results derived from the microanalytic assay of Rahman and Lim [34]. Based on their empirical data, the pricking depth significantly increased with the life stage of *H. halys*. Consequently, the mean depth of penetration measured on tomatoes were between 2.01 and 2.39 mm. It should be noted that the depth of penetration does not match the depth of the lesions formed, as the saliva injected from the rostrum affects deeper tissue layers than the penetration itself. The involvement of the deeper layers (from SR3 to SR8) also supports this notion.

Any plant-related harmful agents, such as abiotic or biotic stressors interfere with the antioxidant system of plants, regardless of whether it is enzymatic or nonenzymatic [35]. Antioxidant capacity has been widely used for the quantification of the overall amount of nonenzymatic antioxidants [36]. The change of absorbance during the measurement is directly related to the combined reducing power of the electron donating antioxidants present in the reaction mixture [26].

Tissue structure changes due to detrimental processes affect the antioxidant capacities of fruit negatively [19] and the antioxidant capacity of tomato has been used as a quality indicator [20], but the information is lacking on the effect of *H. halys* sucking on the antioxidant capacity of tomato fruit. The results indicate that the FRAP values are related to the degradation processes of the mesocarp tissue of tomato fruit caused by the watery saliva released by *H. halys*. To overcome the physical and chemical defenses of plants, the saliva contains enzymes, such as peptidases, glucosidases, and lipases [37]. Lipases are responsible for the degradation of membrane lipids, the primary site of lipid oxidation. The results indicated an initiated, but discontinued lipid oxidation that may be a consequence of enhanced lipase activity in the tomato fruit; however, this process needs further confirmation.

Overall, our noninvasive examination pointed out the serious plant tissue-destructing effect of *H. halys* salivary and its temporal spreading. The pest feeds directly under the exocarp layer but impairment of the placenta containing the seeds and embryo has not occurred. However, the commercial marketability of tomatoes and their use as food will be jeopardized by the injected saliva as time progresses.

After the elaboration of the technical details, the presented experimental method can be suitable for the qualitative control of the vegetable items intended for trade, similar to the procedures developed for other plant products [38,39]. A noninvasive, post-harvest, qualitative control integrated into the technological process can help the isolation of those tomatoes which have suffered bug damage. Ultimately, the widespread application of this method can contribute to the realization of healthy and complete vegetable production.

## 5. Conclusions

The volume change- and tissue-destructive effects caused by H. halys on tomatoes can be excellent analyzed by means of computed tomography. The damaged tomato caused by H. halys has already been examined utilizing several analytical methods, but the noninvasive approaches to investigation assisted by computed tomography has not been applied yet to the exact cognition of the inner enzymatic tissue degression. We quantified the extent of volume alteration and inner structure deviation in tomatoes, which was also confirmed by measured values of antioxidant activity. The escalation of the inner tissue lesions triggered by bug number and the exposure time has been proven by our noninvasive analysis. In summary, the presented experimental method can be suitable for the qualitative control of the vegetable items intended for trade, which can help in the isolation of tomatoes damaged by bugs immediately after harvest.

## Figures and Tables

**Figure 1 biology-11-01070-f001:**
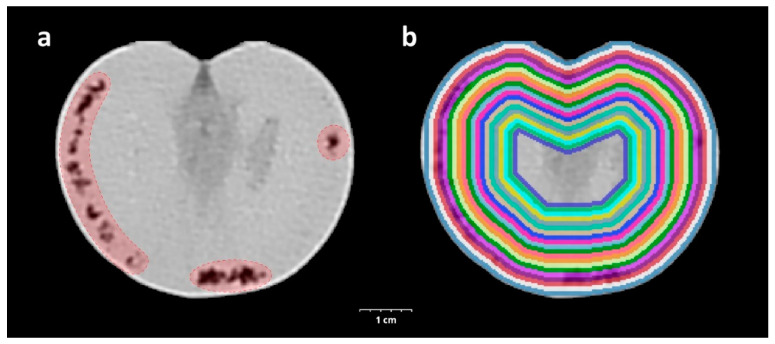
CT-assisted visualization of the triggered inner structure changing of experimental tomato damaged by *H. halys*. The tissue destruction caused by the injected salivary is highlighted in red (**a**). Visualization of the 20 concentric, approximately 1 mm thick binary masks created for sampling shells from the CT scan of a tomato (**b**).

**Figure 2 biology-11-01070-f002:**
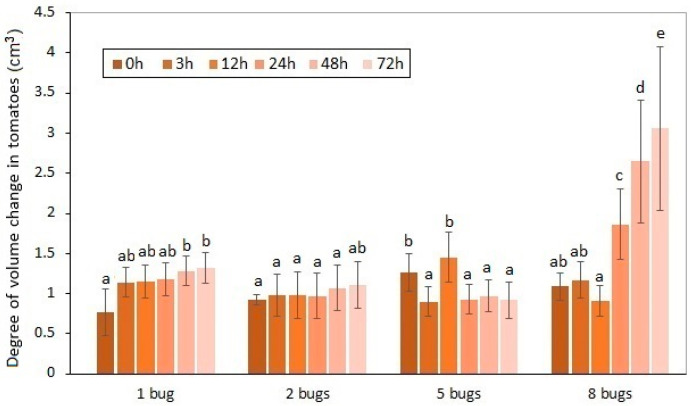
Volume decreasing of experimental tomatoes as a function of exposure time and the number of *H. halys* images. a, b, c, d, e: Small letters indicate a significant difference (*p* ≤ 0.05) between means of different treatments.

**Figure 3 biology-11-01070-f003:**
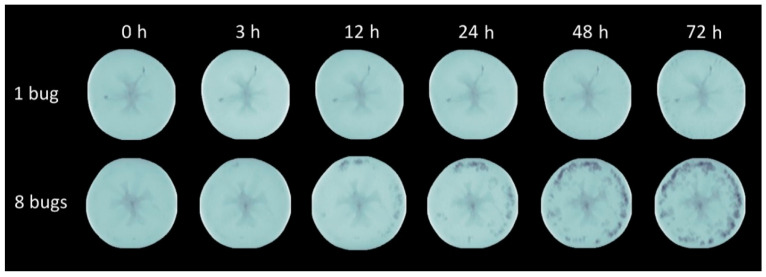
Intensity normalized axial projection.

**Figure 4 biology-11-01070-f004:**
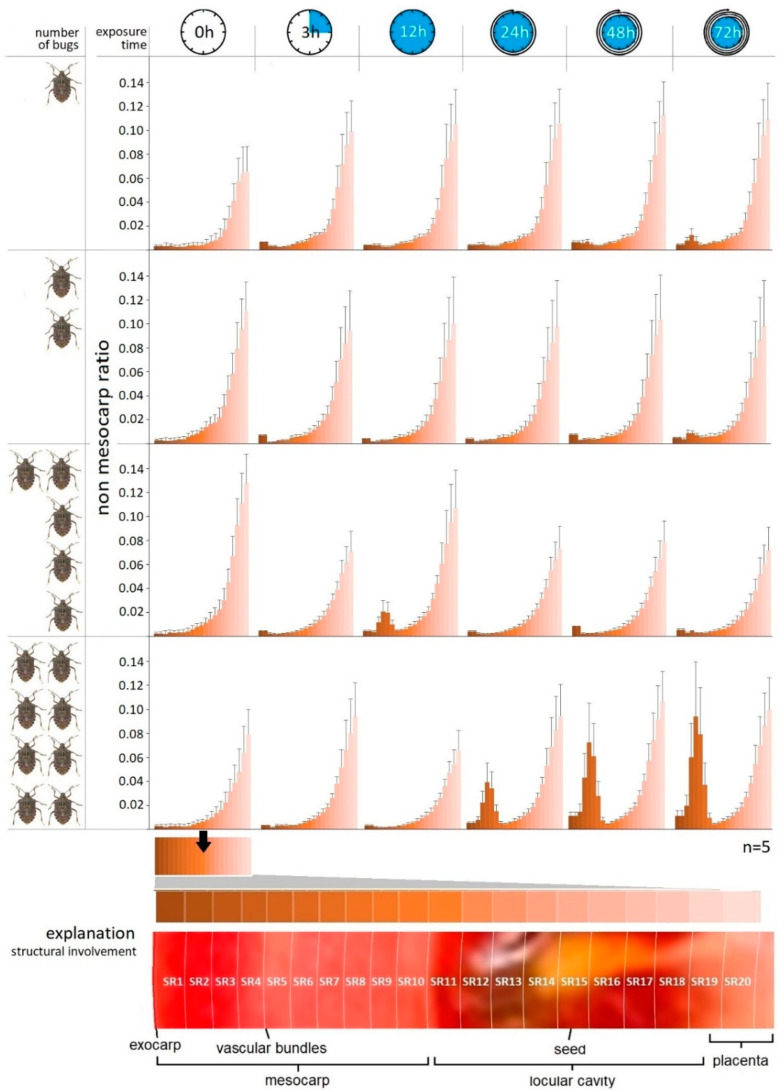
Trends in the tissue structure (SR) in different concentric shells of tomato as a function of exposure time and number of *H. halys* imagoes.

**Figure 5 biology-11-01070-f005:**
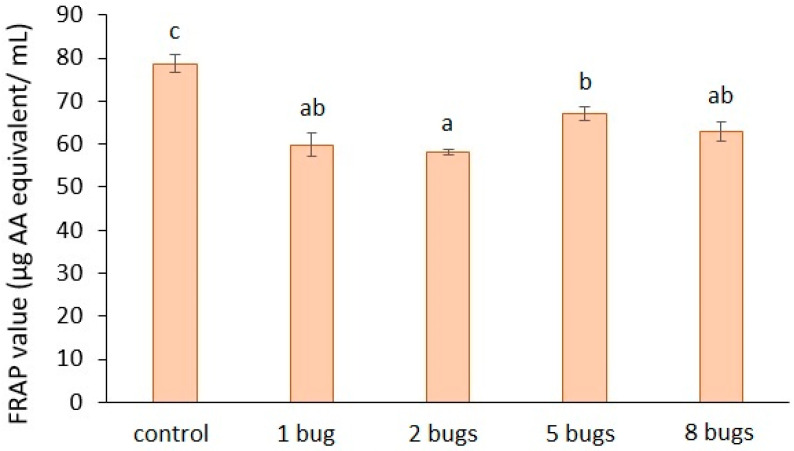
Antioxidant capacity of bug-infested tomato mesocarp. The presented results are the averages of three independent measurements, and the bars represent the ± standard deviation. a, b, c: Small letters indicate significant difference (*p* ≤ 0.05) between the means of different treatments.

## Data Availability

Data are available upon request from the authors.

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
