# Peer review of "Analysis of the Destructive Effect of the Halyomorpha halys Saliva on Tomato by Computer Tomographical Imaging and Antioxidant Capacity Measurement"

_biology, 2022, doi:10.3390/biology11071070_

Round 1

Reviewer 1 Report

Dear Editor and Authors,

I reviewed manuscript biology-1800198 titled, “Analysis of the destructive effect of the Halyomorpha halys saliva on tomato by non-invasive imaging and antioxidant capacity measurement.” The authors conducted laboratory studies linking internal tomato tissue damage caused by H. halys feeding with insect density and exposure time. The methods, results, and conclusions were generally easy to follow. The figures were also informative and presented well. Some of the technical writing could be improved further for succinctness. Provided some revisions are made, this manuscript would be a good addition to the scientific literature concerning the evaluation of insect feeding damage using non-invasive imaging techniques.  

Additional comments and suggested edits below.

Thank you.

L42 Does “(Hemiptera: Pentatomidae)” need to be italicized (?). Please check journal guidelines.

L50 Spelling: polyphagous? Also, the statement of more than 100 host plants should have a reference. Insert reference [13]? Or the following website could be mentioned: http://www.stopbmsb.org/where-is-bmsb/host-plants/

L52 Apple was not listed as one of the affected host plants? H. halys feeding caused more than 37 million USD in apple crop losses in the US. Please see below:

American/Western Fruit Grower2011Brown marmorated stink bug causes $37 million in losses to Mid-Atlantic apple growers. http://www.growingproduce.com/article/21057/brown-marmorated-stink-bug-causes-37-millionin-losses-to-mid-atlantic-apple-growers

L69 Remove terminal S from insects. Insect-caused damage.

L93 Good to see damage-free tomatoes were used as controls.

L120 Thanks for describing the process for creating 20 binary masks.

L153 Suggestion: “Prior to statistical analyses, the normality assumption for data derived from the volume and the tissue analyses (n˃50) was tested using the Shapiro–Wilk test (Ghasemi and Zaherdiasl 2012).” Please add the appropriate reference for Ghasemi and Zaherdiasl to the list of references.

L164 Please add the reference for the statistical software to the list of references.

L168 Remove the word visually.

Figure 2 Y-axis label: Should the first word be capitalized?

All figures in general: Please make sure that the figure axis labels are consistent in text size and font. I noticed differences across the various figures.

Figure 4. For the Y-axis label can you please make the font slightly larger?

Author Response

Dear Reviewer,

thank you for the consideration of our MS, entitled “Analysis of the destructive effect of the Halyomorpha halys saliva on tomato by non-invasive imaging and antioxidant capacity measurement”. We send our revised article on the basis of the opinion of peer-reviewer.

We corrected all remarks, mistakes in MS, which are indicated by colored text by means of track changes.

Corrections are followings, item by item:.

L42 Does “(Hemiptera: Pentatomidae)” need to be italicized (?). Please check journal guidelines.

Thanks a lot for your recommendation, but in our oppinion, that the name of taxonomic order (without species scientific name) need not to written in italic style.

L50 Spelling: polyphagous? Also, the statement of more than 100 host plants should have a reference. Insert reference [13]? Or the following website could be mentioned: http://www.stopbmsb.org/where-is-bmsb/host-plants/

The missing reference is inserted and the its order in the reference list was actualised.

L52 Apple was not listed as one of the affected host plants? H. halys feeding caused more than 37 million USD in apple crop losses in the US. Please see below:

American/Western Fruit Grower. 2011. Brown marmorated stink bug causes $37 million in losses to Mid-Atlantic apple growers. http://www.growingproduce.com/article/21057/brownmarmorated-stink-bug-causes-37-millionin-losses-to-mid-atlantic-apple-growers

 Th missing host is inserted to the list

L69 Remove terminal S from insects. Insect-caused damage.

The word is corrected based on the suggestion.

L153 Suggestion: “Prior to statistical analyses, the normality assumption for data derived from the volume and the tissue analyses (n˃50) was tested using the Shapiro–Wilk test (Ghasemi and Zaherdiasl 2012).” Please add the appropriate reference for Ghasemi and Zaherdiasl to the list of references.

The missing reference is inserted and the its order in the reference list was actualised.

L164 Please add the reference for the statistical software to the list of references.

The missing reference is inserted and the its order in the reference list was actualised

L168 Remove the word visually.

The word is deleted

Figure 2 Y-axis label: Should the first word be capitalized?

It is corrected

All figures in general: Please make sure that the figure axis labels are consistent in text size and font. I noticed differences across the various figures.

All figures and their captions were unified.

Figure 4. For the Y-axis label can you please make the font slightly larger?

The asked labels is enlarged.

We hope the revised MS and our given responses meet the requirements both of the appreciated Reviewer and Biology MDPI.

.. ..

Yours sincerely,

  1. Keszthelyi

corresponding author

Reviewer 2 Report

General 

The main contributions of the paper to the field are summarized as follows:

1. Determine H. halys-induced tissue damage in tomato by non-invasive imaging, in this case Computed Tomography (CT)

2. The proposed technique is applied and tested using a parallell or case-control study with different levels of treatment (1, 2, 5 and 8 bugs). 

3. The findings were validated or confirmed by stress indicating analytical methods such as antioxidant activity.

In general, the paper is well organized, relevant in content and well written. The are some issues and lack of supporting references for non-invasive imaging of plant, CT protocols and how this method could be applied.

1. Introduction

Line 72-74. I am missing some non-invasive imaging techniques which may be more applicable and less expensive, such as ultrasound and RGB or grayscale imaging of surfaces (if possible). Do we need to penetrate the tomato in order to assess the damage in practice?

Line 106-108. There are no reference material used for the CT protocol. How did the authors decide on the current protocol, especially for kV?

Line 112. Could you please elaborate what open-source library used in Python?

In general, I am missing some introduction on how this could be applied or other more practical ways of measuring pest damage in general.

The title says "..by non-invasive imaging..", thereby the authors also needs to look at other more practical and simpler methods, such as ultrasound, to measure pest damage. Maybe a better title would be "..by computed tomography..", since the paper seem to focus mostly on this technique.

Author Response

Dear Reviewer,

thank you for the consideration of our MS, entitled “Analysis of the destructive effect of the Halyomorpha halys saliva on tomato by non-invasive imaging and antioxidant capacity measurement”. We send our revised article on the basis of the opinion of peer-reviewer.

We corrected all remarks, mistakes in MS, which are indicated by colored text by means of track changes.

Corrections are followings, item by item:.  

  1. Introduction

Line 72-74. I am missing some non-invasive imaging techniques which may be more applicable and less expensive, such as ultrasound and RGB or grayscale imaging of surfaces (if possible). Do we need to penetrate the tomato in order to assess the damage in practice?

It is replaced

Line 106-108. There are no reference material used for the CT protocol. How did the authors decide on the current protocol, especially for kV?

In our former studies, good contrast ratios could be achieved in plant parts using a tube voltage of 100kV:

Keszthelyi, S.; Bosnyákné Egri, H.; Horváth, D.; Csóka, Á.; Kovács, Gy.; Donkó, T. (2018) Nutrient content restructuring and CT-measured density, volume attritions on damaged beans caused by Acanthoscelides obtectus Say (Coleoptera: Chrysomelidae) JOURNAL OF PLANT PROTECTION RESEARCH 58 : 1 pp. 91-95. , 5 p. 

Keszthelyi, S. et al., 2021. A Non-Invasive Approach in the Assessment of Stress Phenomena and Impairment Values in Pea Seeds Caused by Pea Weevil. PLANTS-BASEL, 10(7)

Line 112. Could you please elaborate what open-source library used in Python?

DCM images were converted into nifti file format with dcm2niix.

nibabel 3.1.1 - file i/o

numpy 1.21.5 - numerical computations

scipy  1.7.1 - binary morphology, statistics

skimage 0.17.2 - binary morphology

matplotlib - visualization

open-cv2 4.6 – visualization

pandas  1.3.5 - file i/o

In general, I am missing some introduction on how this could be applied or other more practical ways of measuring pest damage in general.

The missing thougths and their related citation is inserted. The reference list is actualised.

The title says "..by non-invasive imaging..", thereby the authors also needs to look at other more practical and simpler methods, such as ultrasound, to measure pest damage. Maybe a better title would be "..by computed tomography..", since the paper seem to focus mostly on this technique.

The title is changed based on the remarks of the Reviewer

We hope the revised MS and our given responses meet the requirements both of the appreciated Reviewer and Biology MDPI.

Yours sincerely,

  1. Keszthelyi

corresponding author

Reviewer 3 Report

line 175, please use % character instead of the word 'percent'

Author Response

Dear Reviewer,

firstly we are very indebted to your valuable contributions to our manuscript, entitled „Analysis of the destructive effect of the Halyomorpha halys saliva on tomato by computer tomographical imaging and antioxidant capacity measurement”. We send our revised article on the basis of the opinion of peer-reviewer.

We corrected the asked changes.

We hope the revised MS and our given responses meet the requirements of Biology MDPI.

Yours sincerely,

  1. Keszthelyi

corresponding author

Reviewer 4 Report

Dear Authors,

I have now completed my review of "Analysis of the destructive effect of the Halyomorpha halys saliva on tomato by non-invasive imaging and antioxidant capacity measurement" (Sándor Keszthelyi, Szilvia Gibicsár, Ildikó Jócsák, Dániel Fajtai and Tamás Donkó) for Biology.

My conclusion is "Accept Submission".

My remarks and suggestions for correction and completion can be seen in the manuscript.

Final English control of the manuscript is necessary.

Author Response

Dear Reviewer,

firstly we are very indebted to your valuable contributions to our manuscript, entitled „Analysis of the destructive effect of the Halyomorpha halys saliva on tomato by computer tomographical imaging and antioxidant capacity measurement”. We send our revised article on the basis of the opinion of peer-reviewer.

We have corrected all asked remarks, and linguistically improved the MS, which can be seen into file with track changes.

We hope the revised MS and our given responses meet the requirements both of our appreciated Reviewer and Biology MDPI.

Yours sincerely,

  1. Keszthelyi

corresponding author
